# High-performance ionomer-free gas diffusion cathodes with low Pt loading for proton exchange membrane water electrolysis

Mingliang Chen [1], Peter M. Piechulla[1], Alexandros Mantzanas[1], Mena-Alexander Kräenbring[2], Fatih Özcan [2,3], Doris Segets [2,3] & J. Ruud van Ommen[1] ✉

Platinum (Pt) is recognized as the most active material for the hydrogen evolution reaction in acidic media; however, its catalytic activity is often underestimated in proton exchange membrane water electrolysis (PEMWE) due to poor utilization of the cathode catalyst layer. In this study, we present the synthesis, characterization, and application of Pt nanoparticles with atomic precision on a microporous-layer-coated gas diffusion layer for PEMWE. The Pt nanoparticles were synthesized via atomic layer deposition, a technique that enables precise control over loading and particle size at the atomic scale. The resulting gas diffusion electrode with an exceptionally low platinum loading (1.08–5.40 µg cm$^{-2}$) demonstrated mass activity at least one order of magnitude higher than that of benchmark Pt. Furthermore, the electrode exhibited exceptional stability at a current density of 1 A cm$^{-2}$ over 200 hours. It also showed robust performance under dynamic operation, enduring 25,000 cycles of alternating cell voltages between 1.45 V and 2 V.

Green hydrogen, produced through water electrolysis (WE) powered by renewable energy, has emerged as a promising solution for both transportation and energy storage, addressing the challenges of global warming and climate change[1]. Among the various WE technologies, proton exchange membrane water electrolysis (PEMWE) is attracting increasing attention due to its high current density operation, high energy efficiency, tolerance for intermittent operation and extended operational lifetime[2].

In PEMWE, the cell is constructed by closely connecting the anode and cathode electrodes through a proton-conductive membrane. However, the acidity of the membrane limits the choice of catalysts to mostly scarce and critical raw materials, specifically iridium (Ir) as the anode catalyst and platinum (Pt) as the cathode catalyst due to their superior activity and stability[3]. The costs and limited availability of these noble metals are likely to become significant bottlenecks when scaling up hydrogen production to levels that could meaningfully contribute to replacing fossil fuels[4].

The sluggish kinetics of the oxygen evolution reaction (OER) at the anode has attracted significant research attention[5–8]. Researchers have investigated various approaches to either increase the activity of Ir further,

e.g. by using hexavalent iridium oxide[6], or replace Ir with alternative materials, such as ruthenium (Ru)[9–11] or cobalt tungstate[12].

On the cathode side, Pt is widely recognized as the most effective catalyst for the hydrogen evolution reaction (HER)[13–15]. Compared to Ir, Pt received considerably less attention[11,12], presumably due to its better availability. Yet, Pt is still a scarce and very expensive metal. Even though efforts have already been dedicated to reducing the Pt loadings[16–18], the catalyst layer (CL) utilization is still low in a membrane electrode assembly (MEA)[13,19].

One recent study has underscored the importance of the triple interface - water, CL, and membrane - in oxygen/hydrogen generation[20]. Observation of bubble formation primarily at the edges of the electrode's pores suggests that a substantial portion of the catalyst does not actively participate in the reaction, leading to lower CL utilization[21,22]. The low CL utilization in an MEA can be partly attributed to the conventional fabrication process, which generally involves three main steps (Fig. 1A): loading the catalyst onto a support material, formulating an ink, and then spray- or slot-die-coating either onto a membrane to form a catalyst-coated membrane (CCM) or onto a gas diffusion layer (GDL) to create a gas diffusion electrode (GDE).

[1]Department of Chemical Engineering, Delft University of Technology, 2629 HZ Delft, The Netherlands. [2]Institute for Energy and Materials Processes – Particle Science and Technology, Universität Duisburg-Essen, 47057 Duisburg, Germany. [3]Center for Nanointegration Duisburg-Essen, Universität Duisburg-Essen, 47057 Duisburg, Germany. ✉e-mail: J.R.vanOmmen@tudelft.nl

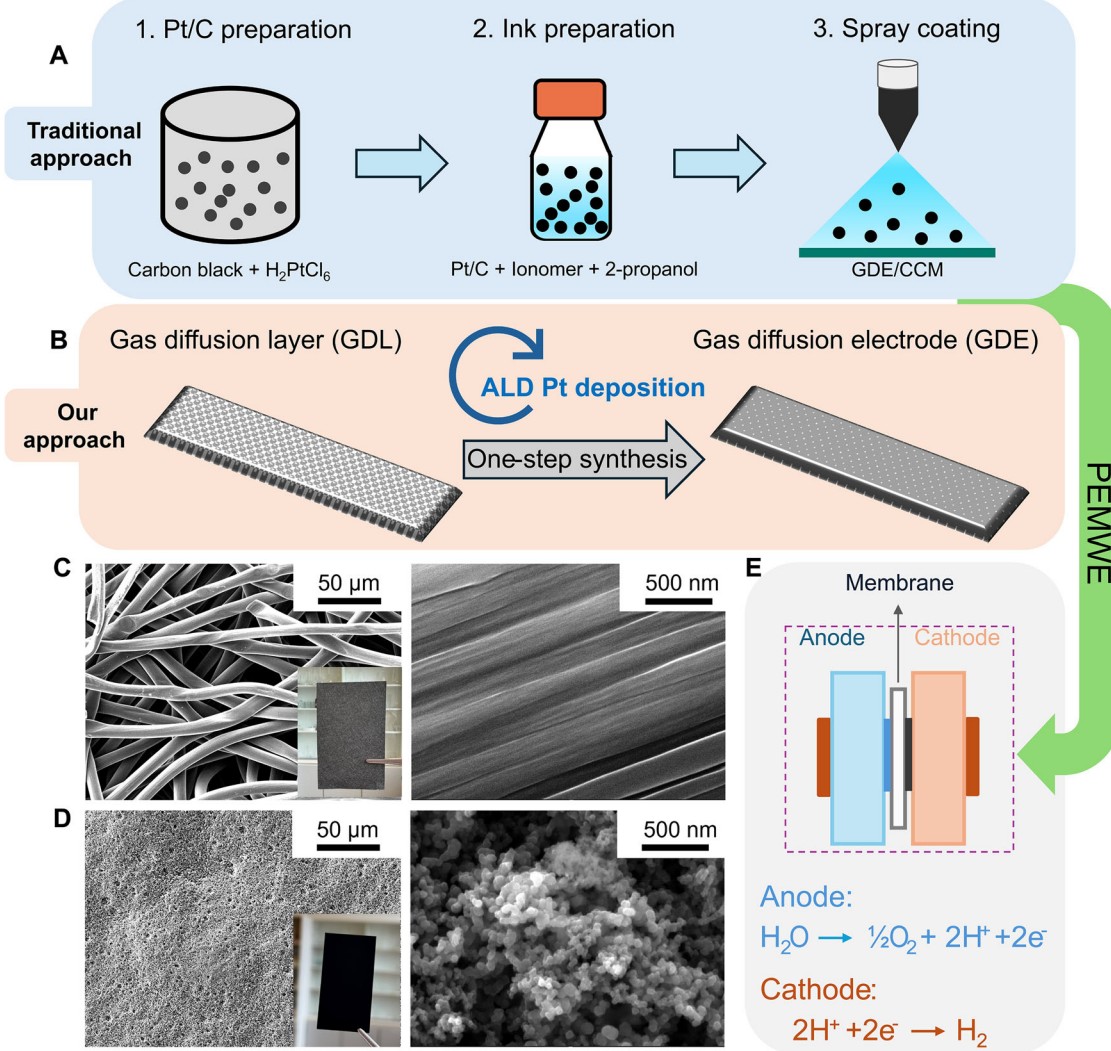

**Fig. 1 | Comparison of traditional catalyst layer fabrication processes (GDE or CCM type) with the ionomer-free GDE fabrication using atomic layer deposition (ALD).** A Traditional three-step fabrication for processing a CCM or GDE, B ionomer-free GDE fabrication using ALD. In this work, commercial GDLs with and without a microporous layer (MPL) were both used as the substrate for ALD Pt deposition. C Scanning electron microscopy (SEM) images of a GDL without an MPL (H23) and (D) SEM images of a GDL with an MPL (H23C6). E Schematic of a proton exchange membrane water electrolyser (PEMWE) for $H_2$ production.

This approach presents several notable challenges with respect to both the processing and design of the CL. The processing aspect includes high labor costs due to multiple fabrication steps and organic solvent waste from ink formulation. With respect to design, the thickness of the CL and the presence of ionomer leads to relatively low CL utilization[23,24]. This has motivated the exploration of more efficient coating techniques (e.g., gas-phase synthesis) combined with reduced CL thickness and catalyst loading (e.g., by using a GDL coated with an ultrathin microporous layer, MPL), and improved control over catalyst particle size[25,26]. However, these activities are mainly limited to the anode side of PEMWE[23,26].

In this work, we demonstrate for the first time a gas-phase synthesis technique - atomic layer deposition (ALD) - for the preparation of ionomer-free PEMWE cathodes. ALD is a self-limiting synthesis technique which can be used to build structures from nanoclusters to thin film growth[27–30]. Due to its precise control of materials at the atomic level, ALD has been adapted to design and prepare catalysts from single atoms to nanoclusters and nanoparticles[31–34]. For instance, the Pt nanoparticles synthesized by ALD on carbon/graphene supports have shown higher mass activity than commercial Pt/C for HER[25,35]. Herein, we report the successful synthesis of Pt nanoparticles with atomic precision on a GDL in one single step using ALD (Fig. 1B). Crucially, Pt deposition on GDLs is highly compatible with spatial

ALD in a roll-to-roll process, which facilitates scalable electrode production[36]. Furthermore, spatial ALD offers enhanced precursor utilization, making it a more cost-effective and competitive fabrication technique compared to conventional spray-coating or slot-die-coating methods.

For the current study, we aim to reduce the loading of Pt through limiting the deposition only on the top surface of the GDL by introducing a plasma source in the ALD deposition process[37]. In addition, we compare the performance of the ALD-made GDEs with and without an MPL (Fig. 1C and D) to a Pt/C GDE and a commercial benchmark CCM in PEMWE (Fig. 1E). The resulting ALD-made GDEs with an MPL demonstrated similar performance to commercial CCMs in terms of cell overpotential, despite an extremely low Pt loading (1.08 ~ 5.40 μg cm$^{-2}$), which is significantly below the future Pt target (50 μg cm$^{-2}$) set by international renewable energy agency (IRENA)[38] and those reported in literature[39–42]. The stability and durability of these ALD GDEs are also evaluated.

## Results and discussion
### Characterization of Pt gas diffusion electrode made by ALD
Prior to the ALD process, the substrate was treated with $O_2$ plasma to activate the surface by increasing the concentration of oxygen functional groups, which has been confirmed by X-ray photoemission spectroscopy

https://doi.org/10.1038/s43246-026-01076-2   **Article**

(XPS) (Supplementary Fig. 1). These oxygen functional groups then facilitated Pt nanoparticle formation on the support materials through alternating doses of MeCpPtMe$_3$ (Pt precursor), O$_2$ plasma (co-reactant) and H$_2$ in sequence within the reactor (see scheme in Supplementary Fig. 2).

A pulse time of 2 s was selected for MeCpPtMe$_3$, as surface saturation was reached at this duration (see XPS loading in Supplementary Fig. 3b), despite slight undersaturation observed in the bulk loading data (see Supplementary Fig. 3a). For the O$_2$ plasma, a pulse time of 30 s was chosen for subsequent experiments, as the saturation curve indicated that Pt loading stabilized at pulse times of 20 s or longer (Supplementary Fig. 3c). The effect of temperature on the Pt deposition was studied by varying the deposition temperature from 100 to 175 °C. The morphology of the carbon fibers and the loading of Pt after deposition were similar to each other except for the one deposited at 100 °C, where large clusters and higher loading were observed (see Supplementary Fig. 4 and 5). The ALD temperature window for Pt was thus determined to be between 125 and 175 °C, with 150 °C selected for all subsequent deposition processes.

Based on these ALD parameters, we investigated the effect of ALD cycles on Pt loading and particle size on carbon fiber supports. The pristine carbon fiber exhibited a smooth surface, which became progressively rougher due to Pt deposition after multiple ALD cycles (see Supplementary Fig. 6). Consistent with previous studies, the particle size on the fiber surface increased with the number of ALD cycles[43,44]. Similarly, Pt surface loading on the carbon fibers increased from 4.85 µg cm$^{-2}$ to 154.47 µg cm$^{-2}$ as the number of ALD cycles increased from 15 to 100. The particle size of Pt was further quantified using transmission electron microscopy (TEM) images (see Supplementary Fig. 7), revealing an increase in the particle diameter from 2.76 ± 0.84 nm (30 ALD cycles) to 16.9 ± 8.6 nm (100 ALD cycles). The deposition of Pt was additionally confirmed by X-ray diffraction (XRD), where characteristic reflexes of Pt at (111), (200), (220) and (311) crystal planes were clearly observed for ALD_N100 (Supplementary Fig. 8).

Next, we applied ALD Pt deposition on a GDL with an MPL (H23C6) using ALD cycles ranging from 5 to 50. The MPL consists of a smooth layer of carbon black with a fluorocarbon binder on top. Deposited Pt nanoparticles on the carbon black were only clearly visible for samples subjected to 20 or more ALD cycles (SEM in Supplementary Fig. 9). TEM imaging was employed to further quantify the particle size of Pt at different ALD cycles. For ALD_N5_MPL, the Pt particles were uniformly dispersed on the carbon black surface, with the majority of particle diameters ranging from 0.5 to 1 nm (Fig. 2A and B). As the number of ALD cycles increased, the particles grew larger, while their homogeneity remained unaffected.

High-resolution XPS was employed to confirm the oxidation states of Pt across various ALD cycles (Fig. 2C). The Pt 4f peaks exhibited symmetric Gaussian-Lorentzian line shapes for both Pt 4f$_{7/2}$ and Pt 4f$_{5/2}$, with each peak deconvoluting into two or three components depending on the number of ALD cycles. For a low number of cycles (ALD_N5_MPL), the Pt 4f$_{7/2}$ peak deconvoluted into two peaks at binding energies of 72.85 eV and 74.29 eV, corresponding to Pt$^{2+}$ and Pt$^{4+}$, respectively. When the ALD cycle count increased to 10 (ALD_N10_MPL), the Pt 4f$_{7/2}$ binding energies were negatively shifted by 0.21 and 0.07 eV due to the increased particle sizes or metal-support interaction[45]. Notably, no metallic Pt was detected in either sample. However, this absence is not expected to impact the electrochemical performance of the electrode, as Pt oxides are also active catalysts for HER[46]. Moreover, as a cathode electrode, Pt oxide can gradually be reduced to metallic Pt during cell operation (Supplementary Fig. 10). When the number of ALD cycles increased to 20 (ALD_N20_MPL), three species of Pt were identified in the sample, including metallic Pt, which accounted for less than 10%, as confirmed by the quantitative analysis of XPS data (Fig. 2D). However, the proportion of metallic Pt increased to over 60% in the ALD_N30_MPL and became the dominant species (>90%) in ALD_N50_MPL, where Pt$^{4+}$ was no longer detected.

The loading of Pt was examined using three different techniques, i.e., inductively coupled plasma mass spectrometry (ICP-MS), energy-dispersive X-ray spectroscopy (EDS), and XPS. All methods demonstrated an increase in Pt loading with an increasing number of ALD cycles

(Fig. 2E). However, the highest Pt loading was observed with XPS, followed by EDS and ICP-MS. This indicates that Pt nanoparticles were predominantly deposited on the surface of the support materials, as XPS is a surface-sensitive technique with a measurement depth typically <10 nm, while ICP-MS provides bulk material measurements (see schematic in Fig. 2F). Additionally, EDS mapping of Pt, C, and F confirmed the homogeneous deposition of Pt across the carbon black surface (Supplementary Fig. 11).

## Impact of Pt loading and the role of MPL on electrochemical performance

To evaluate the electrochemical performance of the ALD GDEs for H$_2$ production in an MEA, three types of MEAs (a commercial benchmark CCM, a commercial Pt/C GDE and ALD GDEs, see scheme in Fig. 3A) were compared using current-polarization curves. For the ALD GDEs, results are presented and discussed in detail for ALD_N5_MPL, ALD_N10_MPL, ALD_N50_MPL, and ALD_N100 in the following sections. Polarization curves of all ALD GDEs with and without an MPL are provided in the supplementary materials (Supplementary Fig. 12 and Fig. 13).

The current polarization curves (Fig. 3B) exhibit two distinct regimes: at low current density, the performance is dominated by catalyst kinetics[20], while at high current density, it is influenced by gas transport, catalyst utilization, and ohmic resistance[47,48]. Due to its lower catalyst utilization, the cell voltage of ALD_N100 (154 µg/cm²) was higher than that of the commercial Pt/C GDE at low current density. This is evidenced by the Tafel slope of 71.49 mV dec$^{-1}$ for the ALD_N100 GDE, compared to 49.41 mV dec$^{-1}$ for the Pt/C GDE (Fig. 3C). However, in the high current density regime, ALD_N100 outperformed the Pt/C GDE. Across all current density regimes, ALD_N100 showed inferior performance than the commercial benchmark CCM which benefits from a much higher catalyst loading.

Adding an MPL to the carbon fiber surface significantly enhances PEMWE performance[26,49–51]. The cell voltage of ALD_N50_MPL was much lower than that of the commercial benchmark CCM, despite having more than 16 times lower Pt loading. With a further reduction in Pt loading to 5.40 µg cm$^{-2}$, ALD_N10_MPL still demonstrated a slightly lower cell voltage than the commercial CCM at high current density. However, the performance of ALD_N5_MPL (1.08 µg cm$^{-2}$) was slightly worse than that of the commercial CCM but remained comparable to ALD_N100, even with five times less Pt loading. Tafel analysis reveals a slope of 52.03 mV dec$^{-1}$ for ALD_N50_MPL vs. 51.37 mV dec$^{-1}$ for the commercial CCM, indicating similar HER kinetics despite the significantly lower Pt loading of ALD_N50_MPL (61.0 µg cm$^{-2}$ vs. 1.0 mg cm$^{-2}$). The Tafel slope of ALD GDEs (Fig. 3C) with an MPL increased slightly as the Pt loading decreased, explaining the observed performance trends.

To better understand the differences in cell performance, a breakdown analysis of cell voltage was conducted. After subtracting the potential contribution from high-frequency resistance (HFR), the commercial CCM exhibited the best performance among the samples (Fig. 3D). We attribute the HFR (Supplementary Fig. 14) observed in the commercial CCM to the poor contact between CL and carbon fiber (in-plane resistance) or a thick CL (through-plane resistance), leading to elevated electrical resistance. At high current densities, the Pt/C GDE demonstrated poorer performance than the other samples. As already pointed out in the Tafel analysis, ALD_N100 exhibited a much higher kinetic overpotential (approx. +60 mV, Fig. 3E) compared to the other samples due to lower catalyst utilization. The differences among the ALD-made GDEs with an MPL, the commercial CCM, and the Pt/C GDE were much smaller, i.e., within 10 mV range from each other, which aligns with their similar Tafel slopes. Regarding transport overpotential (Fig. 3F), the Pt/C GDE exhibited significantly higher values at high current densities, contributing to its elevated cell voltage. This high transport resistance was further confirmed by electrochemical impedance spectroscopy (EIS) analysis, which showed two semi-arcs in the Nyquist plot, with the second arc being larger than the first (Supplementary Fig. 14)[52]. In contrast, the transport overpotential for ALD GDEs with an MPL was higher than that of the commercial CCM and

   **3**

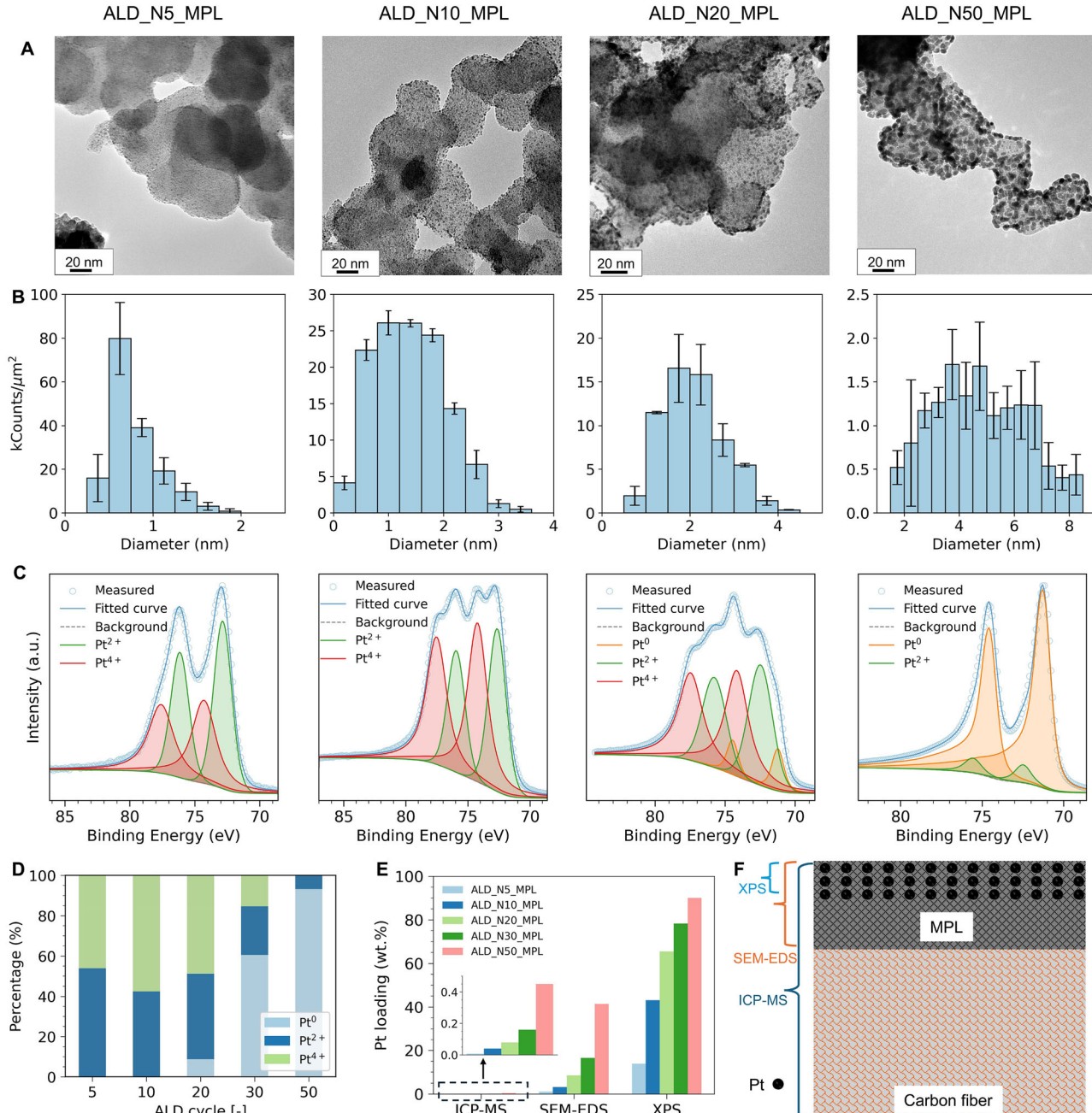

**Fig. 2 | Characterization of Pt gas diffusion electrode. A** TEM images of Pt-coated GDEs and (**B**) particle size distribution of Pt made by ALD at different numbers of cycles based on a count of over 300 particles (the error bars represent the deviation of particle sizes within defined bin widths), **C** Pt 4f XPS spectra of Pt-coated GDEs, Pt 4f peak deconvolutes into two or three peaks in the samples: Pt⁰, Pt²⁺ or Pt⁴⁺. **D** Percentage of each peak in the Pt-coated GDEs, **E** Pt loading of the Pt-coated GDEs measured by ICP-MS, SEM-EDS, and XPS. **F** Schematic illustration of the Pt deposition area on the MPL-coated GDL, and the detection depth of the used techniques. Source data for Fig. 2 are provided in Supplementary Data 1.

ALD_N100 but remained similar across these samples due to their comparable cell structures. Both the commercial CCM and ALD_N100 demonstrated low mass transport resistance, as reflected in their single semi-arc EIS curves (Supplementary Fig. 13).

The role of the MPL in improving PEMWE performance has been well discussed in previous work[26,49–51,53]. However, here we demonstrate how the Pt loading can be minimized by combining the ALD technique with an MPL-coated GDL while maintaining excellent electrode performance in a PEMWE. This is because the smaller pores and particle sizes of the MPL contribute to a smoother surface, leading to more homogeneous surface contact between catalyst nanoparticles and membrane[50,53]. This, in turn, significantly enhances CL utilization and improves electrode performance

(see schematic in Fig. 4A). Additionally, a smooth interface between transport layer and membrane also facilitates the use of thinner membranes, thereby further reducing overpotentials by reducing the ohmic contribution[54].

Compared to the CCM configuration, GDEs offer advantages such as better contact between the CL and the GDL, as well as reduced deformation of the CL due to membrane swelling[23]. However, spray-coated Pt/C GDEs often form thick CLs with ink penetration into the GDL, reducing catalyst particle accessibility[50]. In contrast, the plasma-enhanced ALD technique allows for the coating of Pt nanoparticles directly on the MPL surface, minimizing penetration into the substrate. ALD therefore enables deposition of the catalyst exactly at the membrane-MPL interface, and with precise

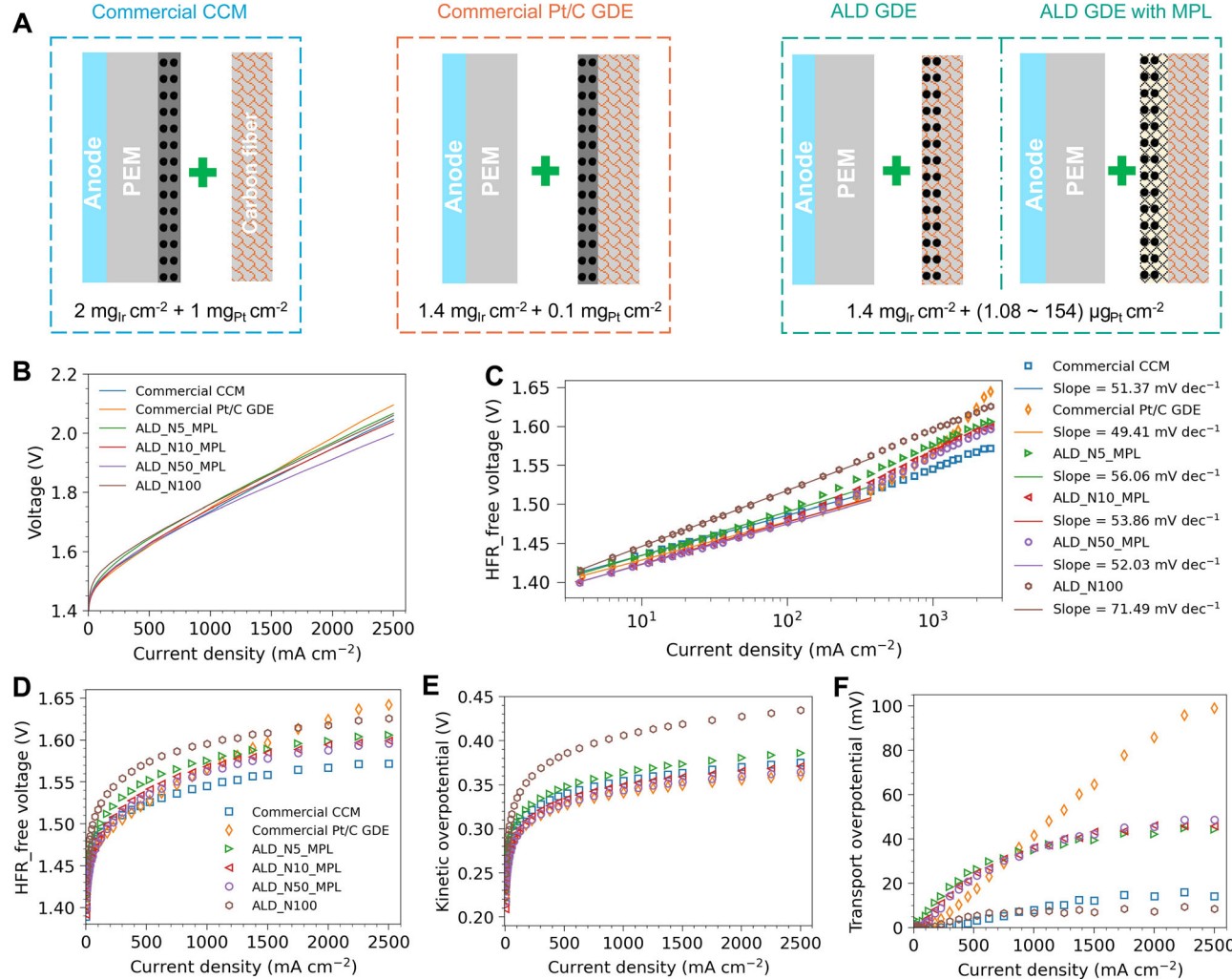

**Fig. 3 | Three different types of MEAs are compared in terms of current polarization curves. A** Schematic illustration of a commercial CCM configuration, a spray-coated cathode GDE with commercial Pt/C and ALD cathode GDEs with and without an MPL. For all cathode GDE configurations, a commercial single-side coated CCM was used on the anode side (1.4 mg$_{Ir}$ cm$^{-2}$). **B** Electrochemical performance comparison of ALD GDEs, the commercial CCM and the spray-coated GDE. **C** Tafel slope analysis of all samples. The breakdown analysis of the cell voltage is composed of HFR-free voltage (**D**), kinetic overpotential (**E**) and transport overpotential (**F**) (see methods for detailed explanation). The ohmic resistance of the cell is determined by EIS analysis. Source data for Fig. 3 are provided in Supplementary Data 2.

control over particle size and homogeneity, thereby maximizing CL utilization[20].

The superior performance of ALD GDEs can also be attributed to the hydrophobic treatment of the MPL with a polytetrafluoroethylene (PTFE) agent provided by the manufacturer. This hydrophobicity remains intact after ALD treatment, as confirmed by the strong fluorine signal detected in both SEM-EDS and XPS analyses (Supplementary Figs. 11 and 15). The hydrophobic surface of the MPL facilitates improved mass transport properties, in particular the removal of H$_2$ bubbles, by balancing the liquid and gas phases within the electrode network[55,56].

To compare the mass activity of our ALD GDEs with commercial CCM and Pt/C GDE, we normalized the exchange current density to the catalyst mass (Fig. 4B). For ALD GDEs with an MPL, the mass activity increased as the number of ALD cycles decreased, attributed to the smaller particle size and higher electrochemical surface area of the catalyst (Supplementary Fig. 16). Remarkably, all ALD GDEs demonstrated higher mass activity compared to both commercial CCM and Pt/C GDE. In particular, the mass activity of ALD_N5_MPL was over three orders of magnitude higher than the commercial CCM and two orders of magnitude higher than the Pt/C GDE. Encouraged by these results, we further benchmarked the electrochemical performance of ALD GDEs

against state-of-the-art CCMs and GDEs reported in the literature (Fig. 4B and Supplementary Table 1). Surprisingly, the specific activity of ALD_N5_MPL and ALD_N10_MPL surpassed all previously reported values, while the latter does not compromise performance in absolute terms (see polarization curve discussion above). This highlights the transformative potential of combining ALD with MPL-coated GDLs to fabricate electrodes with exceptionally low Pt loadings (~99.5% reduction (5.40 µg cm$^{-2}$) per GW of H$_2$ compared to commercial CCMs (1 mg cm$^{-2}$)) for future PEMWE applications.

## Long-term stability during PEM electrolysis

To evaluate the stability of the ALD GDE with an MPL during PEM electrolysis, we have conducted a chronopotentiometry stability test (CST) with ALD_N10_MPL at 80 °C for 200 hours under a current density of 1 A cm$^{-2}$. The PEMWE cell showed stable performance with the cell voltage stabilizing at approximately 1.75 V throughout the test period, despite a minor degradation rate of 0.069 mV h$^{-1}$ (Fig. 5A). Stability was further confirmed through current polarization measurements taken before and after CST (Fig. 5B), which showed nearly identical curves with differences below 3.53 mV at 1 A cm$^{-2}$, indicating consistent cell performance.

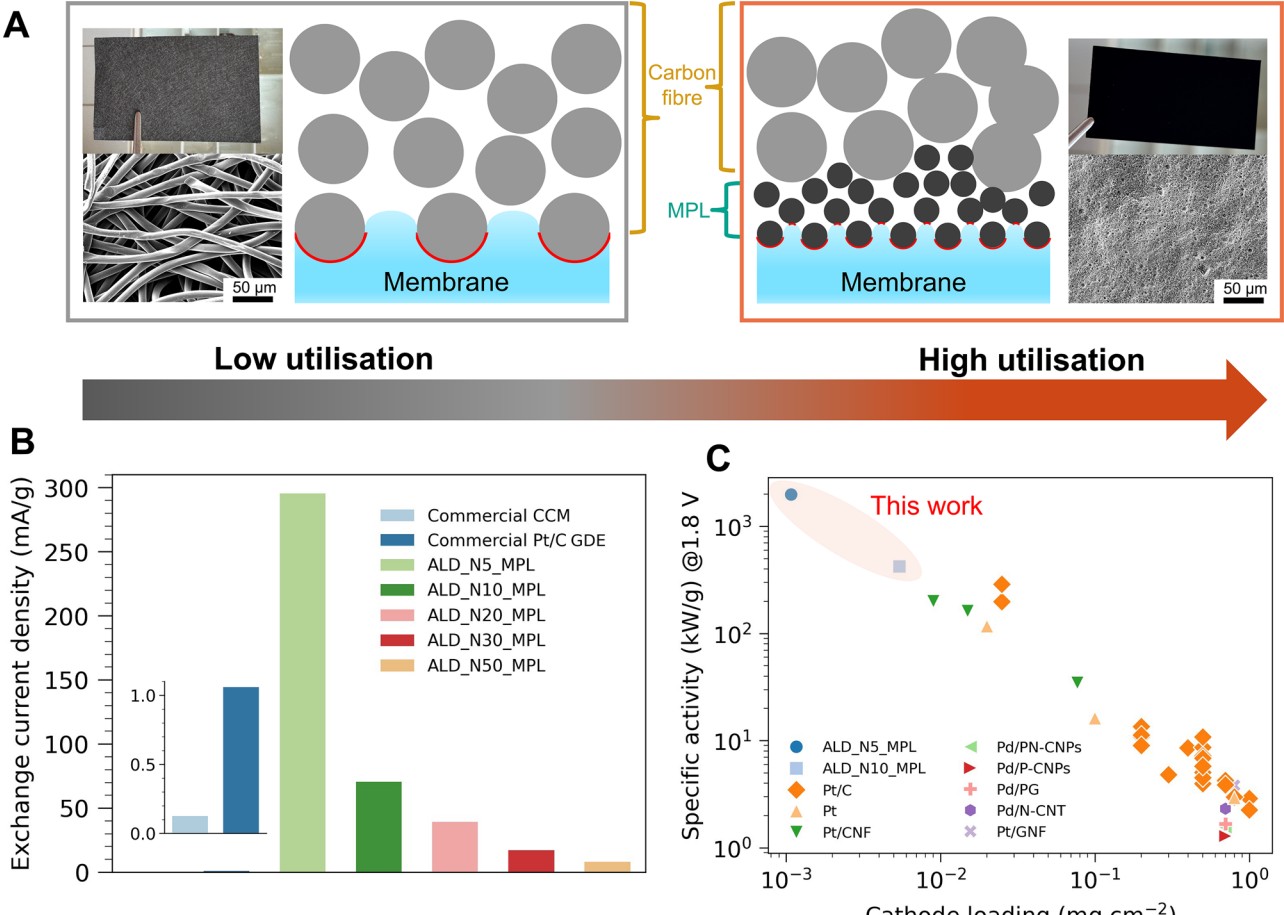

**Fig. 4 | Catalyst layer utilization and performance comparison with those from literature. A** Schematic illustration of catalyst layer utilization of GDEs with and without an MPL (adapted with permission from Ref. 53). b Comparison of mass normalized exchanged current density of ALD GDEs, the commercial CCM and the commercial Pt/C GDE. The exchange current density is calculated by the Butler-Volmer equation. **C** Performance comparison of ALD GDEs with those state-of-the-art CCMs and GDEs from literature. A complete table of the literature is given in the supplementary information. To ensure a fair comparison, only those with an anode catalyst loading above 1 mg cm$^{-2}$ were chosen. Source data for Fig. 4 are provided in Supplementary Data 3.

EIS was performed at different current densities before and after CST to gain deeper insights into electrode stability (Fig. 5C and D). Two circuit fitting models[52] were developed to analyze the EIS data under low and high current density conditions, respectively (Fig. 5E). The high current density model included mass transport contributions, as evidenced by two semi-arcs in the EIS curves. Fitting the models revealed that the ohmic resistance ($R_s$) and charge transfer resistance ($R_{ct}$) of the cell remained unchanged before and after CST (Fig. 5F). These results confirm that the cell and catalyst activity were unaffected during the test.

To further evaluate the durability of the cell, we subjected ALD_N30_MPL to an accelerated stress test (AST) at 80 °C following a protocol adapted from literature (Fig. 6A)[57]. The test involved alternating the cell voltage between 1.45 V and 2 V, holding each voltage for 5 s per cycle, for a total of 25k cycles, representing at least 600 hours of operation using intermittent renewable sources such as wind or solar[57]. Current polarization curves were recorded every 5k cycles, showing a difference below 12 mV at 1 A cm$^{-2}$ after 25k cycles (Fig. 6B).

Additionally, EIS data (Fig. 6C) were collected at the same intervals and analyzed using the circuit model shown in Fig. 5E to derive the ohmic resistance ($R_s$) and charge transfer resistance ($R_{ct}$). The $R_s$ remained stable at approximately 0.156 Ω*cm$^2$, while $R_{ct}$ exhibited only a slight increase from 0.251 Ω*cm$^2$ to 0.252 Ω*cm$^2$ (Fig. 6D). This minimal degradation may be attributed to CL reconstruction, such as nanoparticle aggregation (Supplementary Fig. 17), and/or the partial dissolution of noble metals. Overall,

our direct ALD-made cathodes with low Pt loading showed high durability in a dynamic operating condition.

## Conclusions

The reported strategy offers a straightforward approach for developing highly stable ionomer-free cathode electrodes with exceptional performance and extremely low Pt loading (~99.5% reduction per GW of H$_2$ compared to commercial CCMs) for PEMWE applications. By integrating ALD with MPL-coated GDLs, the synthesized electrodes achieve specific mass activities an order of magnitude higher than previously reported results in PEMWE cells, while maintaining the same performance as the commercial reference in terms of cell overpotentials.

Looking ahead, the presented Pt ALD process can be transferred to a spatial ALD process in a roll-to-roll setup, which offers enhanced precursor utilization[58]. ALD-based electrode manufacturing therefore presents itself as a scalable and cost-effective fabrication technique compared to conventional spray-coating or slot-die-coating methods, paving the way for next-generation, high-performance electrodes with low Pt loadings for industrial hydrogen production applications.

## Methods
### Chemicals and materials

Commercial double-sided CCMs, anode single-sided CCMs, and GDLs with (H23C6, Freudenberg Performance Materials, Germany) and

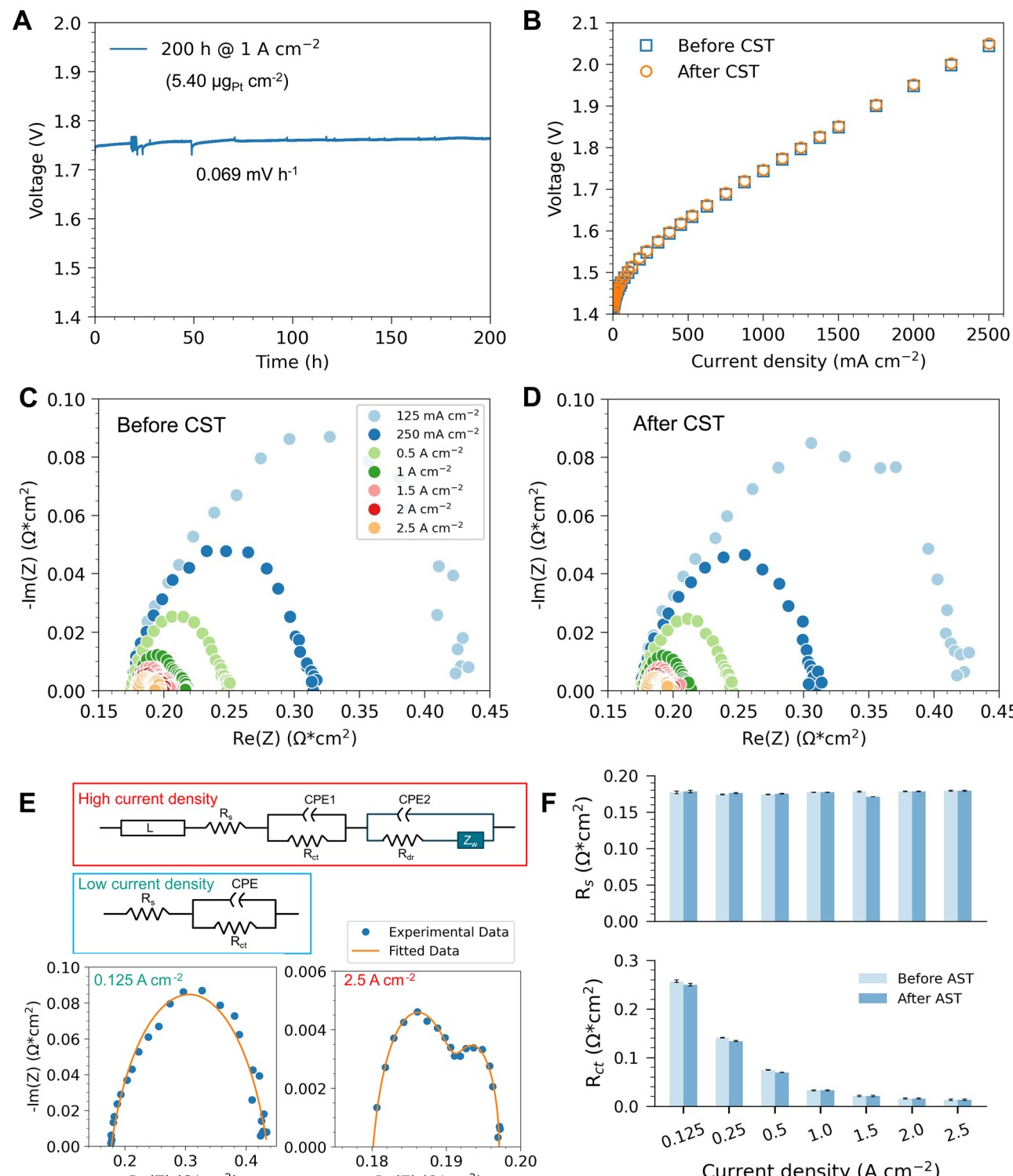

**Fig. 5 | Long-term stability. A** Long-term stability of the MEA cell with a cathode of ALD_N10_MPL running at 1 A cm$^{-2}$ for 200 h. **B** Current polarization curve of the cell before and after chronopotentiometry stability test (CST). EIS analysis of the cell before CST (**C**) and after CST (**D**). **E** Fitting models for EIS at low current density and high current density. **F** Ohmic resistance ($R_s$) and charge transfer resistance ($R_{ct}$) of the cell before and after CST. At high current density, the EIS is composed of two semi-arcs and the one at lower frequency can be ascribed to the mass transport limitation. Source data for Fig. 5 are provided in Supplementary Data 4.

without (H23) an MPL were purchased from QuinTech (Germany). For both double-sided and single-sided CCMs, Nafion$^{TM}$ N117 was used as the proton conductive membrane. The anode and cathode loading of double-sided CCM are 2 mg$_{Ir}$ cm$^{-2}$ and 1 mg$_{Pt}$ cm$^{-2}$, respectively. The anode loading for a single-sided CCM is 1.4 mg$_{Ir}$ cm$^{-2}$. The active surface area of the commercial CCMs is 5 cm$^2$ (2.23 cm × 2.23 cm).

(Trimethyl)methylcyclopentadienylplatinum(IV), MeCpPtMe$_3$ (99%, CAS: 94442-22-5) was ordered from Strem. O$_2$ (electronic grade 99.999 + %), H$_2$ (electronic grade 99.999 + %) and ultrapure Ar were used in the ALD reactor. Isopropyl alcohol (≥99.9 %, CAS: 67-63-0) was purchased from Honeywell. Commercial Pt/C (10 wt% Pt/Vulcan XC72, catalog number 738581) and H$_2$SO$_4$ (95-97%) were ordered from

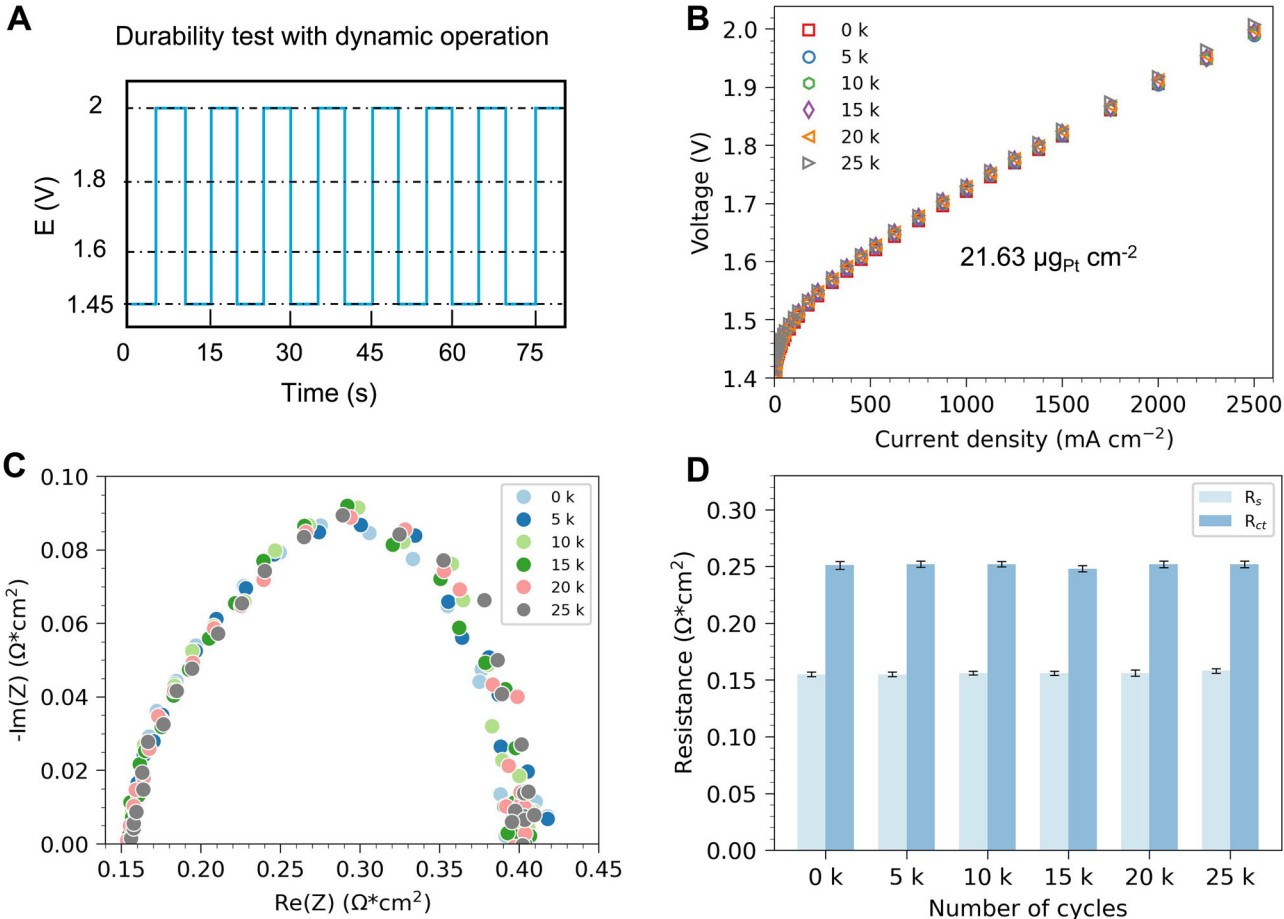

**Fig. 6 | Durability test with dynamic operation. A** Schematic of durability test with dynamic operation. During this dynamic operation, the cell voltage alternated between 1.45 V and 2 V, holding each voltage for 5 s per cycle. The total running cycle is 25k, and the current polarization and EIS were analyzed every 5k cycles. **B** The current polarization of the cell measured every 5k cycles of accelerated stability test (AST). **C** EIS analysis of the cell after different cycles of durability test at a current density of 125 mA cm$^{-2}$. **D** Ohmic resistance ($R_s$) and charge transfer resistance ($R_{ct}$) of the cell after different cycles of durability test. Source data for Fig. 6 are provided in Supplementary Data 5.

Sigma-Aldrich. Unless otherwise stated, deionized water was used for the experiments.

## Atomic layer deposition of Pt on GDL

The deposition of Pt on GDL was conducted in a commercial ALD reactor (Fiji, Veeco, USA). To investigate the deposition quality of Pt on the carbon fiber surface, we used a GDL without an MPL (H23) as the support to determine the optimal pulse times of precursors and the ALD temperature window. The MeCpPtMe$_3$ precursor was heated to 75 °C, and the gases were used at room temperature. The O$_2$ and H$_2$ flow rates were set to 50 and 20 sccm, respectively. Each ALD cycle consisted of three sequential pulses: (a) MeCpPtMe$_3$ precursor dosing, (b) O$_2$ plasma exposure and (c) H$_2$ exposure. The H$_2$ pulse was introduced to reduce the PtO$_x$ and mitigate the growth delay of the deposition during the nucleation stage[43]. The MeCpPtMe$_3$ was introduced into the deposition chamber using Ar as carrier gas. A 10 s purge was applied after the MeCpPtMe$_3$ pulse to ensure that the reaction byproduct and excessive precursors were removed from the chamber. To identify optimal deposition conditions, the pulse time for MeCpPtMe$_3$ was varied between 0.25 s and 6 s, while the exposure time of O$_2$ plasma was adjusted from 10 s to 30 s. The H$_2$ exposure time was fixed at 10 s. The reactor chamber temperature was varied between 100 °C and 175 °C to determine the optimal deposition temperature. The ALD samples are labeled as ALD_Nx or ALD_Nx_MPL, where x represents the number of ALD cycles.

## Cathode electrode preparation using spray-coating method with commercial Pt/C

Catalyst inks were prepared by dispersing 0.5 wt.% solids in a solvent mixture of 50 vol.% isopropanol and 50 vol.% deionized water (18.2 MΩ cm) using an Ohaus Explorer EX125 analytical balance with an accuracy of 0.1 mg. The solid components consisted of 75 wt.% Pt/C (with a Pt loading of 10 %) and 25 wt.% Nafion D2020. To ensure uniform dispersion, the mixture was first tip sonicated (Branson SFX 550) for 30 seconds (20 % amplitude), followed by bath sonication (Elmasonic S30H) for 20 minutes, and then tip sonicated again for 30 seconds at the same amplitude, following an optimized procedure[59]. All dispersion procedures were conducted in ice to avoid excessive heat and solvent evaporation. The heating and dispersion characteristics of the tip sonicator at this amplitude were previously determined[60].

After dispersion, the ink was loaded into a PRISM 400 Ultra-Coat ultrasonic spray coater by USI, Inc. and sprayed using an ink flow rate of 0.4 ml min$^{-1}$, a nozzle speed of 250 mm s$^{-1}$, a nozzle distance of 40 mm, and a coating area of 7 × 7 cm$^2$ at a temperature of 100 °C on a vacuum hot plate. The catalyst loading was then determined gravimetrically using a Kern ABP 200-5DM analytical balance with an accuracy of 0.1 mg.

## Characterization

XPS measurements were conducted by ThermoFisher Kα system with a photon energy of 1486.7 eV. The system was maintained at high vacuum condition during measurement (10$^{-9}$ mbar). Survey scan was performed to

identify the elemental composition of the sample, while high-resolution scans were applied to some specific elements to identify the chemical states.

The morphology of the samples was observed by field emission scanning electron microscopy (FE-SEM) (JSM-IT700HR, JEOL, Japan) at an accelerating voltage of 5 kV, a spot size of 30, and a working distance of approximately 10 mm, respectively. EDS was performed to identify the atomic percentage of each element present on the sample surface at an accelerating voltage of 10 kV, a spot size of 60, and a working distance of approximately 10 mm, respectively. EDS mapping was carried out to obtain the nanostructure information of the sample.

The nanoparticle size of the sample was analyzed by TEM using a JEOL JEM1400 (Japan) at a voltage of 120 kV. The Pt/C sample was dispersed with isopropyl alcohol and water respectively and then dropped on a Cu200 grid (Quantifoil, Germany). For the Pt-coated GDL sample, a cutter knife was used to scratch the powder off from the Pt coated side of the GDL and then followed the same procedure for the sample preparation. ImageJ was employed to get the particle sizes from the TEM images counting more than 300 particles.

XRD measurements were carried out to analyze the crystal structure of the sample using a Bruker D2 Phaser instrument (USA). It was operated with Cu Kα irradiation at 40 kV and 25 mA. The XRD pattern of the sample was acquired in the 2θ range of 10-90° with a measuring time of 0.1 s per step.

The Pt loading on ALD-deposited GDLs was analyzed using ICP-MS using Nexion 2000 system (USA). Approximately 30 mg sample was destructed in 1.5 ml 65% $HNO_3$ and 4.5 ml 30% HCl using a microwave. The destruction time in the microwave was 60 min at max power. After destruction, the sample was diluted to 50 ml with Milli-Q water. The samples were then diluted 100 times in 1% $HNO_3$ prior to analysis.

## Electrochemical test

A home-made test station equipped with a potentiostat (VSP 300, BioLogic, France) and booster (10 A, BioLogic, France) was used to study the CCM and GDE configurations. Prior to use, both the commercial single-sided anode CCMs and the commercial double-sided CCMs were treated with 1 M $H_2SO_4$ and deionized water at 80 °C for 1 h, respectively. For the commercial CCM setup, the cell assembly followed this sequence: an anode gasket (1 mm PTFE), Ti felt (1 mm), CCM, cathode gasket (0.1 mm PTFE), and carbon fiber (H23). The components were stacked together and secured between a Ti anode end plate and a stainless-steel cathode end plate (Dioxide Materials, FL, USA). The assembly was fastened with eight M5 screws tightened to 5 Nm, which were inserted into the end plates. Both end plates featured a central flow field with a 5 cm$^2$ parallel finger structure to ensure uniform water distribution across the GDLs. The effective working area of the cell was set as 4 cm$^2$ by using 2 × 2 cm GDLs or GDEs, and gaskets with corresponding cut-outs. The Ti felt was cut using a laser, while the carbon fiber was manually cut with a knife. For the GDE configuration (ALD-deposited Pt or spray-coated Pt/C), the assembly procedure was identical, except that a commercial single-sided anode CCM was used.

Prior to the electrochemical characterization, the cell was heated to 80 °C and the deionized water was pre-heated to 80 °C before supplying separately to both electrodes using a peristaltic pump (Minipuls 3, Gilson, The Netherlands). The flow rate of water was set to 25 ml min$^{-1}$ to remove the generated bubbles effectively. The MEA was first conditioned at a current density of 4 A cm$^{-2}$ for at least 6 hours to stabilize the performance. The polarization curve (voltage) was then recorded at various current densities (0.25, 1.25, 2.5, 5, 7.5, 10, 12.5, 15, 17.5, 20, 25, 30, 35, 45, 55, 75, 100, 125, 175, 225, 300, 375, 450, 525, 625, 750, 875, 1000, 1125, 1250, 1375, 1500, 1750, 2000, 2250, 2500 mA cm$^{-2}$), with each current density holding for 2 min. Afterwards, galvanostatic EIS analysis was carried out at seven different current densities (0.125, 0.25, 0.5, 1, 1.5, 2, 2.5 A/cm$^2$) using a frequency ranging from 100 kHz to 100 mHz. The EIS analysis was used to obtain the ohmic resistance of the cell and to identify the possible limiting factors among the MEAs.

For the CST, the cell was running at a constant current density of 1 A cm$^{-2}$ for 200 hours. The cell voltage was recorded accordingly. The current polarization curve and the EIS were measured before and after the stability test. The AST was conducted by running the cell at two different voltages (1.45 V and 2 V) alternatively, holding each voltage for 5 s per cycle with a total running cycle of 25k. The current polarization curve, and the EIS of the cell were measured every 5k cycle. Due to the increase in conductivity of water, the water was refreshed in the middle of the measurement.

The electrochemical surface area (ECSA) of the GDE was measured in a standard three-electrode cell at room temperature. A platinum plate served as the counter electrode, and an Ag/AgCl electrode was used as the reference. The GDEs were cut into 1 × 2 cm$^2$ pieces, with a 1 × 0.5 cm$^2$ section secured by an electrode clamp. The clamp was positioned 0.5 cm above the electrolyte to avoid direct contact. Consequently, only 1 cm$^2$ of the GDE was exposed to the electrolyte and served as the working electrode. Prior to analysis, the working electrode was cleaned by cycling the potential between 0.05 and 1.2 V (vs. RHE) for 20 cycles in 0.5 M $H_2SO_4$ that had been purged with Ar for at least 20 min.

## Breakdown analysis of cell voltage

The cell voltage $E_{cell}$ consists of the following four elements: reversible cell voltage $E^0(p, T)$, kinetic overpotential $\eta_{kin}$, ohmic overpotential $\eta_{ohmic}$, and mass transport overpotential $\eta_{mt}$[23]:

$$E_{cell} = E^0(p, T) + \eta_{ohmic} + \eta_{kin} + \eta_{mt} \tag{1}$$

The high frequency resistance (HFR) or ohmic resistance $\eta_{ohmic}$ of the cell can be determined by EIS analysis, then the cell voltage can be corrected by the ohmic overpotential to obtain the HFR-free cell voltage enabling a comparison among different MEAs unbiased by the setup or membrane resistances. The reversible cell voltage $E^0(p, T)$ can be described by the Nernst equation[61]:

$$E^0(p, T) = 1.2291(V) - 0.0008456(V/K) * (T - 298.15(K)) \tag{2}$$

where $T$ is the cell temperature (K).

To determine the kinetic overpotential $\eta_{kin}$, Tafel slope was first calculated by fitting the HFR-free cell voltage in the current density range from 5 to 100 mA/cm$^2$. Afterwards, $\eta_{kin}$ can be derived from the Butler-Volmer equation[62]:

$$\eta_{kin} = b * \frac{i}{i_0} \tag{3}$$

where $b$ is the Tafel slope (V dec$^{-1}$), $i$ is the current density of the cell, and $i_0$ is the apparent exchange current density. With these parameters defined, the mass transport overpotential $\eta_{mt}$ can be derived by subtracting the reversible cell voltage $E^0(p, T)$, kinetic overpotential $\eta_{kin}$, and ohmic overpotential $\eta_{ohmic}$ from the cell voltage $E_{cell}$.

## Data availability

All data are available in the main text or the supplementary materials. Source data are provided with this paper.

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

## Acknowledgements

We thank Joost Middelkoop for ALD training, Duco Bosma for SEM training and analysis, Wiel Evers for TEM training, Doğa Özerk for TEM analysis on the sample after durability test, Bart Boshuizen for XPS training and analysis, Cas Veenhoven for his assistance in the lab work, Baukje Terpstra for ICP-MS analysis, and Min Li and Ming Li for the helpful discussion on ALD and PEMWE. This research was funded by the ECCM KICkstart DE-NL project (KICH1.ED04.20.023), which is supported by the Dutch Research Council (NWO).

## Author contributions

J.R.v.O. proposed and supervised the project. M.C. designed and conducted the experiments, data analysis of experiments, and illustration. A.M. conducted part of the experiments. P.M.P. and M.A.K. designed and conducted part of the experiments. F.O. and D.S. supervised the project. All authors discussed the results and commented on the manuscript.

## Competing interests

The authors declare no competing interests.
