## [Transparent Peer Review file · Communications Materials]

High-performing ionomer-free gas diffusion cathodes with low loading for proton exchange membrane water electrolysis

Corresponding Author: Professor Ruud van Ommen

Version 0:

Decision Letter:

**** Please ensure you delete the link to your author homepage in this email if you wish to forward it to your coauthors ****

Dear Professor van Ommen,

Thank you for submitting your manuscript, "High-performing ionomer-free gas diffusion cathodes with ultralow loading for proton exchange membrane water electrolysis", to Communications Materials. It has now been seen by 2 referees, whose comments are appended below. You will see that while they find your work of potential interest, they raise substantial concerns that must be addressed. In light of these comments, we cannot accept the manuscript for publication but are interested in considering a revised version that addresses these serious concerns.

In particular, while both reviewers recognize the novelty and promise of your approach in achieving ultralow platinum loading with high activity and stability, they have raised issues regarding the adequacy of experimental evidence and the need for clearer mechanistic understanding to substantiate the reported performance claims.

We hope you will find the referees' comments useful as you decide how to proceed. Should additional data or analysis allow you to address these criticisms, we would be happy to look at a revised manuscript.

To allow us to move forward with your work, we also ask that you edit your manuscript according to the attached table.

Please read this document carefully as we will be unable to further assess your revised paper until these important points are addressed.

Please outline all revisions made in the right-hand column and return the completed table with your updated manuscript files as a Related Manuscript file.

When resubmitting, please also include:

- A point-by-point response to the reviewers' comments. If you are unable to address specific reviewer requests or find any points invalid, please explain
- A clean version of your revised manuscript with no mark-ups
- A marked-up version of your paper with all changes highlighted in a different colour

Please use the link below to submit your revised files:

Link Redacted

**** This url links to your confidential home page and associated information about manuscripts you may have submitted or that you are reviewing for us. If you wish to forward this email to co-authors, please delete the link to your homepage first ****

We hope to receive your revised paper within twelve weeks, but we understand that revisions may take longer. Please let us know if you find that the revision process will take substantially more time.

We are committed to providing a fair and constructive peer-review process. Please do not hesitate to contact me if you have any questions or would like to discuss these revisions further. We look forward to seeing the revised manuscript and thank you for the opportunity to review your work.

Best regards,

Lling-Lling Tan, PhD
Editorial Board Member
Communications Materials
orcid.org/0000-0002-3766-8413

Reviewers' comments:

Reviewer #1 (Remarks to the Author):

The author reported that they have achieved ionomer free low platinum loading between 1.08–5.40 $\mu\text{g}/\text{cm}^2$ on cathode gas diffusion electrode using atomic layer deposition technique for PEM water electrolysis which demonstrated a mass activity of one order of magnitude higher than that of benchmark Pt. Furthermore, the electrode exhibited stability up to 200 hours at a current density of 1 A/cm²

It also showed stable performance up to 25,000 cycles at alternating cell voltages between 1.45 V and 2 V.

Xiao et al., Nature Catalysis (2022) , Ding et al., Nano-Micro Letters / NM Lett (2023) and Ostroverkh et al., Int. J. Hydrogen Energy / ECS work (2017–2019), etc., reported up to 10–50 $\mu\text{g}/\text{cm}^2$ of platinum loading using sputtering/atomic layer deposition techniques which are applied in fuel cells and PEM electrolysis.

since very few publications in this area I recommend this manuscript for publication.

Reviewer #2 (Remarks to the Author):

In this study, the authors present a novel approach for fabricating ionomer-free cathodes for Proton Exchange Membrane Water Electrolysis (PEMWE). The method involves the direct, single-step synthesis of Platinum (Pt) nanoparticles onto a microporous-layer (MPL)-coated gas diffusion layer (GDL) via Atomic Layer Deposition (ALD). The key finding is that this ALD-fabricated electrode achieves exceptional performance despite an ultralow Pt loading (1.08–5.40 $\mu\text{g}/\text{cm}^2$). Notably, the electrode exhibits a mass activity at least one order of magnitude higher than the benchmark Pt catalyst. Furthermore, the authors demonstrate robust stability, with the electrode sustaining operation at a current density of 1 A/cm² for over 200 hours and enduring 25,000 cycles of dynamic operation.

This manuscript presents findings of significant novelty and potential impact for the PEMWE field.

However, to fully substantiate its exceptional claims (ultralow Pt loading, >10x mass activity, high stability) and meet the rigorous standards of Communication Materials, the following major revisions are essential. If the authors can provide this critical characterization data, the revised manuscript will be a strong candidate for publication.

1. The assertions made in the manuscript regarding the remarkably high 'mass activity' and 'catalyst utilization' lack strong evidence in the absence of ECSA data (e.g., obtained from H-UPD). The authors are required to present results from ECSA testing to accurately quantify the genuine electrochemically active surface area. This is the most significant point of concern.
2. The impressive 25,000-cycle stability data is supported only electrochemically. Physical evidence (e.g., post-test TEM/SEM) is required to confirm the morphological integrity of the Pt nanoparticles and definitively rule out degradation mechanisms such as particle aggregation or detachment.
3. The authors must address the apparent contradiction wherein the synthesis method uses an H₂ pulse (ostensibly to reduce PtOx), yet the XPS data for low-cycle samples (e.g., ALD_N5_MPL) shows no metallic Pt₀. A clearer discussion of the actual role of the H₂ pulse at low loadings is needed.

** See Nature Research's author and referees' website at www.nature.com/authors for information about policies, services and author benefits

Version 1:

Decision Letter:

** Please ensure you delete the link to your author homepage in this email if you wish to forward it to your coauthors **

Dear Professor van Ommen,

Thank you once again for submitting your manuscript, "High-performing ionomer-free gas diffusion cathodes with low loading for proton exchange membrane water electrolysis," to Communications Materials. The concerns of our reviewers have been addressed, but there are some amendments needed before we can accept your paper.

We ask that you edit your manuscript according to the attached table. **Please read this document carefully as we will be unable to further assess your revised paper until these important points are addressed.**

Please outline all revisions made in the right-hand column and return the completed table with your updated manuscript files as a Related Manuscript file.

Please use the link below to submit your revised files:
Link Redacted

When resubmitting, please provide a marked-up manuscript with all changes highlighted, as well as a clean version of your paper.

We hope to receive this updated version of your paper within 1 week, but please let us know if you find that you need more time.

Best regards,

Lling-Lling Tan, PhD
Editorial Board Member
Communications Materials
orcid.org/0000-0002-3766-8413

Reviewers' comments:

Reviewer #2 (Remarks to the Author):

The author responds to the comment very well and explains clearly. This work is ready for publication.

Version 2:

Decision Letter:

Dear Professor van Ommen,

We are delighted to accept your manuscript titled "High-performing ionomer-free gas diffusion cathodes with low loading for proton exchange membrane water electrolysis" for publication in Communications Materials. Thank you for choosing to publish your interesting work with us.

Please note that in advance of your paper being published we will host an early access version, known as an 'Article in Press,' on our journal website. For more information on this initiative please see our [author guidelines](https://support.springernature.com/en/support/solutions/articles/6000281821-what-is-an-article-in-press-).

Licence to Publish and Article-Processing Charge

In approximately 7-10 business days you will receive an email with a link to choose the grant of rights necessary for publishing your paper and – if applicable – to provide payment information for your article-processing charge (APC), either via credit card or by requesting an invoice.

If needed, our Author Services team will be in touch regarding any additional information that may be required.

In order to avoid any delays, please ensure that you have emails from Springer Nature whitelisted in your mail system.

We will edit your manuscript to ensure that it conforms with our house style and send you a link to an online eProof for checking in a separate email to the publishing agreements. Please read your proof with great care to ensure that the sense has not been altered. We also suggest you discuss the proof with your co-authors, but please ensure that only one author communicates with us and that only one set of corrections is returned via the online correction in the eProof. The corresponding (or nominated) author is responsible on behalf of all co-authors for the accuracy of all content, including spelling of names and current affiliations.

To ensure prompt publication, your proofs should be returned within two working days. If there is any period within the next four weeks in which you won't be available, please nominate a co-author with whom we can correspond, and let us know their e-mail address as soon as possible.

Please note that production will not continue until the Licence to Publish and Article-Processing Charge steps are completed and your proof corrections are submitted.

Please note that your Supplementary Information files are now finalized. They will be uploaded directly to the Communications Materials website in preparation for publication of the Article. Any requests to make changes will only be considered in exceptional circumstances and will result in a delay to publication.

Acceptance of your manuscript is conditional on all authors' agreement with [our publication policies](https://www.nature.com/commsmat/editorial-policies). In particular, your manuscript must not be published elsewhere and there must be no announcement of the work in the media until the publication date. At this stage, you may wish to make your institution's press office aware of the forthcoming publication, if you wish to bring your work to the media's attention, so that they can start preparing any publicity. Please note that the paper is still under embargo until it is published in the journal. Further details of our embargo policy can be found here <http://www.nature.com/authors/policies/embargo.html>.

We will aim to publish your article in a timely manner. Please note there will be no further correspondence about your publication date. When your article is published, you will receive a notification email. **If you are planning an embargoed press release or require a specific publication date, please complete our [scheduling requests form](https://forms.office.com/e/ed7NBDDd08u), or contact commsproduction@springernature.com, as soon as possible after acceptance and we will endeavour to accommodate your request.** For further information on the journey of your article from acceptance to publication, please see our [Author FAQs](https://www.nature.com/documents/Author_FAQs.pdf).

If you have any questions about open-access invoicing or payment, please contact authororders@nature.com

Best regards,

Lling-Lling Tan, PhD
Editorial Board Member
Communications Materials
orcid.org/0000-0002-3766-8413

***As a new journal, we would greatly appreciate any comments you have about your experience at Communications Materials. I hope that we have been able to meet your expectations and look forward to working with you again in the future.

We may promote your article on social media once it is published, so please feel free to send me the twitter handles of any authors or departments and we will be sure to tag them accordingly.***

Reviewers' comments:

Reviewer #1 (Remarks to the Author):

The author reported that they have achieved ionomer free low platinum loading between 1.08–5.40 $\mu\text{g}/\text{cm}^2$ on cathode gas diffusion electrode using atomic layer deposition technique for PEM water electrolysis which demonstrated a mass activity of one order of magnitude higher than that of benchmark Pt. Furthermore, the electrode exhibited stability up to 200 hours at a current density of 1 A/cm^2

It also showed stable performance up to 25,000 cycles at alternating cell voltages between 1.45 V and 2 V.

Xiao et al., Nature Catalysis (2022) , Ding et al., Nano-Micro Letters / NM Lett (2023) and Ostroverkh et al., Int. J. Hydrogen Energy / ECS work (2017–2019), etc., reported up to 10–50 $\mu\text{g}/\text{cm}^2$ of platinum loading using sputtering/atomic layer deposition techniques which are applied in fuel cells and PEM electrolysis.

since very few publications in this area I recommend this manuscript for publication.

Response: we thank you reviewer 1 for the positive comments on our manuscript. Some relevant references are cited in the revised manuscript.

We made the following revision in the updated manuscript on page 4.

“The resulting ALD-made GDEs with an MPL demonstrated similar performance to commercial CCMs in terms of cell overpotential, despite an extremely low Pt loading (1.08~5.40 $\mu\text{g}/\text{cm}^2$), which is significantly below the future Pt target (50 $\mu\text{g}/\text{cm}^2$) set by international renewable energy agency (IRENA)³⁷ and those reported in literature^{38, 39, 40, 41}.”

Reviewer #2 (Remarks to the Author):

In this study, the authors present a novel approach for fabricating ionomer-free cathodes for Proton Exchange Membrane Water Electrolysis (PEMWE). The method involves the direct, single-step synthesis of Platinum (Pt) nanoparticles onto a microporous-layer (MPL)-coated gas diffusion layer (GDL) via Atomic Layer Deposition (ALD). The key finding is that this ALD-fabricated electrode achieves exceptional performance despite an ultralow Pt loading (1.08–5.40 mg/cm^2).

Notably, the electrode exhibits a mass activity at least one order of magnitude higher than the benchmark Pt catalyst. Furthermore, the authors demonstrate robust stability, with the electrode sustaining operation at a current density of 1 A/cm² for over 200 hours and enduring 25,000 cycles of dynamic operation.

This manuscript presents findings of significant novelty and potential impact for the PEMWE field.

However, to fully substantiate its exceptional claims (ultralow Pt loading, >10x mass activity, high stability) and meet the rigorous standards of Communication Materials, the following major revisions are essential. If the authors can provide this critical characterization data, the revised manuscript will be a strong candidate for publication.

1. The assertions made in the manuscript regarding the remarkably high 'mass activity' and 'catalyst utilization' lack strong evidence in the absence of ECSA data (e.g., obtained from H-UPD). The authors are required to present results from ECSA testing to accurately quantify the genuine electrochemically active surface area. This is the most significant point of concern.

Response: we thank you for the suggestion of reviewer 2 and have added the data on ECSA of ALD GDE samples. This measurement was carried out in a standard three-electrode cell with a platinum plate as a counter and an Ag/AgCl electrode as a reference. The details of this test were added in the Methods section, and the results are provided in the supplementary materials (Figure S16). The ECSA of the samples indicated that the Pt nanoparticles deposited at a lower number of ALD cycles have a higher active surface area.

We added the following information in the updated manuscript on page 20.

“The electrochemical surface area (ECSA) of the GDE was measured in a standard three-electrode cell at room temperature. A platinum plate served as the counter electrode, and an Ag/AgCl electrode was used as the reference. The GDEs were cut into 1 × 2 cm² pieces, with a 1 × 0.5 cm² section secured by an electrode clamp. The clamp was positioned 0.5 cm above the electrolyte to avoid direct contact. Consequently, only 1 cm² of the GDE was exposed to the electrolyte and served as the working electrode. Prior to analysis, the working electrode was cleaned by cycling the potential between 0.05 and 1.2 V (vs. RHE) for 20 cycles in 0.5 M H₂SO₄ that had been purged with Ar

for at least 20 min.”

Fig. S16. Electrochemical surface area of ALD GDEs on H23C6. (a) ALD_N5_MPL, (b) ALD_N30_MPL, (c) ALD_N50_MPL, (d) ALD_N100_MPL.

2. The impressive 25,000-cycle stability data is supported only electrochemically. Physical evidence (e.g., post-test TEM/SEM) is required to confirm the morphological integrity of the Pt nanoparticles and definitively rule out degradation mechanisms such as particle aggregation or detachment.

Response: We examined the particle size of the sample after the 25,000-cycle stability test using TEM (see image below). However, only a small number of Pt nanoparticles remained on the GDE after cell compression and the durability test at 80 °C, as most were likely transferred to the membrane. This is attributed to the fact that Pt particles generated by the plasma-enhanced ALD process are located only on the top surface of the GDL. Based on the TEM image, particle aggregation may have occurred during the durability test, as some larger particles were observed.

Fig. S17. TEM analysis of the GDE after 25,000-cycle durability test. Due to the cell compression and the durability test at 80 °C, most of Pt particles were likely transferred to the membrane, and only a small number of them remained on the GDE.

“This minimal degradation may be attributed to CL reconstruction, such as nanoparticle aggregation (Supplementary Fig. 17), and/or the partial dissolution of noble metals.”

3. The authors must address the apparent contradiction wherein the synthesis method uses an H₂ pulse (ostensibly to reduce PtO_x), yet the XPS data for low-cycle samples (e.g., ALD_N5_MPL) shows no metallic Pt⁰. A clearer discussion of the actual role of the H₂ pulse at low loadings is needed.

Response: In the ALD process, oxygen plasma was applied to remove the ligands of the MeCpPtMe₃ precursor. However, due to the strong oxidizing power of plasma, this step can lead to the oxidation of Pt. To address this, an H₂ pulse was introduced into the

ALD cycle to reduce the surface metal oxide layer back to metallic Pt (Mackus et al. *Chemistry of Materials*, 25(9), 1769-1774). For samples with a low number of ALD cycles, the H₂ reduction step is not sufficiently effective to fully convert the oxide layer into Pt. As the number of ALD cycles increases, a greater amount of metallic Pt can be detected, since the oxidized Pt species are confined mainly to the outer surface of the particles. Using H₂ plasma or extending the H₂ pulse duration could enhance the reduction of PtO_x to Pt. Nevertheless, this was not further investigated, as both PtO_x and Pt are known to be efficient catalysts for the hydrogen evolution reaction (HER). Moreover, during cathodic operation, PtO_x is expected to be reduced to metallic Pt, as confirmed by XPS analysis shown in Fig. S10 of the Supplementary Materials.

Mackus, A. J., Garcia-Alonso, D., Knoops, H. C., Bol, A. A., & Kessels, W. M. (2013). Room-temperature atomic layer deposition of platinum. *Chemistry of Materials*, 25(9), 1769-1774.